# Transporters in the Mammary Gland—Contribution to Presence of Nutrients and Drugs into Milk

**DOI:** 10.3390/nu11102372

**Published:** 2019-10-05

**Authors:** Alba M. García-Lino, Indira Álvarez-Fernández, Esther Blanco-Paniagua, Gracia Merino, Ana I. Álvarez

**Affiliations:** 1Department of Biomedical Sciences, Physiology, Veterinary Faculty, Universidad de León, Campus de Vegazana, 24071 León, Spain; agarl@unileon.es (A.M.G.-L.); eblap@unileon.es (E.B.-P.); gmerp@unileon.es (G.M.); 2Institute of Animal Health (INDEGSAL), Universidad de León, Campus de Vegazana, 24071 León, Spain

**Keywords:** ABC-transporters, lactation, mammary gland, milk, SLC-transporters

## Abstract

A large number of nutrients and bioactive ingredients found in milk play an important role in the nourishment of breast-fed infants and dairy consumers. Some of these ingredients include physiologically relevant compounds such as vitamins, peptides, neuroactive compounds and hormones. Conversely, milk may contain substances—drugs, pesticides, carcinogens, environmental pollutants—which have undesirable effects on health. The transfer of these compounds into milk is unavoidably linked to the function of transport proteins. Expression of transporters belonging to the ATP-binding cassette (ABC-) and Solute Carrier (SLC-) superfamilies varies with the lactation stages of the mammary gland. In particular, Organic Anion Transporting Polypeptides 1A2 (OATP1A2) and 2B1 (OATP2B1), Organic Cation Transporter 1 (OCT1), Novel Organic Cation Transporter 1 (OCTN1), Concentrative Nucleoside Transporters 1, 2 and 3 (CNT1, CNT2 and CNT3), Peptide Transporter 2 (PEPT2), Sodium-dependent Vitamin C Transporter 2 (SVCT2), Multidrug Resistance-associated Protein 5 (ABCC5) and Breast Cancer Resistance Protein (ABCG2) are highly induced during lactation. This review will focus on these transporters overexpressed during lactation and their role in the transfer of products into the milk, including both beneficial and harmful compounds. Furthermore, additional factors, such as regulation, polymorphisms or drug-drug interactions will be described.

## 1. Introduction

Milk is a complete food, one of the main sources of nutrients and bioactive ingredients in mammals and consequently it plays an important role in health. Its consumption is especially relevant in the early stages of the development of newborns.

Mature human milk is a complex emulsion of fat and aqueous fluid containing water and approximately 3.5% proteins, 7% sugars, 4% lipids, 0.5% minerals of total volume [1,2,3]. In recent years, the importance of immunomodulatory processes linked to breast milk components has increased [4,5,6,7]. Nowadays, the presence of immune active molecules, lactoferrin, oligosaccharides, lysozyme and enzymes with antibacterial activities which protect gastrointestinal tract and mammary gland is widely known [8,9]. New discoveries highlight the presence of extracellular vesicles and exosomes involved in cell communication and the regulation of immune processes following milk ingestion by the newborn [10,11,12,13].

Milk production is a complex process unavoidably linked to transport mechanisms [14]. Previous excellent reviews have focused on the study of transporters present in the mammary gland [15,16]. RNA levels in lactating mammary epithelial cells (MEC) purified from pooled fresh milk samples compared with those in non-lactating MEC have revealed that the milk transfer of compounds is mainly mediated by two transporter superfamilies: ATP-binding cassette (ABC-) and Solute Carrier (SLC-). Moreover, their expression varies with lactation stages of the mammary gland [17,18,19,20,21,22,23].

Alcorn *et al.* [18] observed differences in the expression of ABC- and SLC- transporters between lactating human MEC and non-lactating MEC. Indeed, 4-fold higher RNA levels were found for Organic Cation Transporter 1 (OCT1), Novel Organic Cation Transporter 1 (OCTN1), Concentrative Nucleoside Transporters 1 (CNT1) and 3 (CNT3) and Peptide Transporter 2 (PEPT2). Increased transcripts (2.2-fold higher RNA levels) were also detected in lactating MEC for Sodium-dependent Vitamin C Transporter 2 (SVCT2). Finally, mRNA levels for Organic Anion Transporting Polypeptides 1A2 (OATP1A2) and 2B1 (OATP2B1) and Multidrug Resistance-associated Protein 5 (ABCC5/MRP5) were slightly higher (about 1.5-fold) in lactating than in non-lactating MEC. Even though authors attributed to normal physiological variation or interindividual differences such minor changes, they did not discard that the role of these transporters could have an impact on milk composition. Regarding Equilibrative Nucleoside Transporter 3 (ENT3), although Alcorn et al. [18] found differences between lactating and non-lactating MEC, Gilchrist et al. [21] showed that its expression decreased during lactation. Moreover, studies conducted on lactating rat mammary gland and isolated Mammary Epithelial Organoids (MEO) revealed an increased expression of Oct1, Octn1, Cnt1, Cnt2, Cnt3, Pept2 and Svct2 compared to their respective non-lactating controls [21]. Immunohistochemistry and Western blot analysis of mammary gland showed that murine, bovine and human Breast Cancer Resistance Protein (BCRP/ABCG2) was strongly induced during lactation [19]. Lindner et al. [24] obtained similar results confirming that protein expression of ABCG2 was increased in mammary gland from lactating compared with non-lactating cows, sheep and goats.

These proteins are localized in the basolateral or the apical membrane of the mammary epithelium, participating in the uptake, re-uptake or efflux of nutrients and compounds of a different nature, thus contributing to milk composition (Figure 1). The concentration of some of these compounds in the milk, such as vitamins, is especially relevant for newborns during lactation, since milk is their only source of nutrients [25]. Consequently, some studies have reported high mortality rates as well as severe neurological and motor disorders in children who were fed with formulas deficient in thiamine [26,27]. Adults who suffer from cow’s milk allergy may also be at risk of vitamin deficiency [28]. Conversely, these transporters feature broad substrate specificities and they mediate active transport of toxic chemicals, such as drugs, pesticides, carcinogens and environmental pollutants into milk [23,29,30]. In fact, most ABC and SLC transporters are involved in the detoxification and elimination of xenobiotics potentially harmful for the organism. Therefore, the expression of these transporters in the intestine, the liver, the kidney or the placental, hematotesticular and blood-brain barriers, constitutes a defence system [31]. Their activity in the mammary gland, however, involves a deeper and more complex interpretation. On the one hand, these transporters play a beneficial role in contributing to the transfer of nutrients into milk, which is in contrast with the secretion of harmful compounds which can contaminate milk [32]. This feature represents a major concern for Public Health and Food Quality and Safety, since both newborns and dairy product consumers may be exposed to these dangerous compounds. A deeper understanding of the transport processes in the lactating mammary gland is crucial for study design and protection of women and their infants. Moreover, the exposure to contaminants as well as the administration of veterinary drugs in other food-producing animals, such as poultry or swine, may also imply a risk to consumers of products of animal origin different from milk. Understanding the activity of SLC and ABC transporters present in these animal species is essential to predict the presence of toxic residues in products such as meat or eggs. In this regard, Schrickx and Fink-Gremmels [33] and Virkel et al. [34] recently reviewed the role of ABC transporters in the bioavailability and toxicology of veterinary drugs in different species, including swine and horses.

There are many drugs of human and veterinary use that can be transferred into milk if they are administered during lactation. Considerable efforts are being made to predict the presence of drugs in milk based on pharmacokinetic parameters [16,47]. Furthermore, milk residues of toxins or chemicals can alter the expression of transporters which, in turn, mediate the transference of these undesirable compounds into milk [30]. Milk composition strongly affects xenobiotic uptake and concentration. Protein binding could play a role in drug transfer into milk. Although some proteins present in the milk, such as casein, lactoferrin and albumin have the ability to bind drugs, their impact on milk concentration of such drugs does not appear to be relevant [48]. However, fat content is one of the main factors that contribute to the concentration of hydrophobic drugs into milk [49]. Considering the important changes produced in milk composition during the different stages of lactation [50] and also, that milk composition varies between species [51], these factors must be taken into account to predict xenobiotic transfer into milk.

Exposure through milk intake to unprescribed medicines may have adverse effects, including drug-drug interactions leading to decreased therapeutic effect, enhanced side effects of co-administered drugs or altered disposition of endogenous and dietary compounds [52]. Moreover, the unintentional intake of some drugs in sensitive people may trigger allergic responses. Antibiotics and antiparasitic drugs are of special concern since continued exposure to these compounds may lead to the development of drug resistances. To avoid the risk of drug exposure through ruminant milk, the administration of veterinary medicines is strictly regulated by institutional authorities such as the European Food Safety Authority (EFSA) in Europe or the U.S. Food and Drug Administration (FDA) in the United States. Concern for these risks has encouraged the FDA to develop a multi-criteria ranking model for the risk management of drug residues in milk. Based on their extensive use, likelihood of appearing in milk, probability of human exposure and hazardous consequences of exposure, the drugs that gave the highest score were beta-lactam antibiotics, antiparasitics, macrolides, aminoglycosides, non-steroidal anti-inflammatory drugs (NSAIDs), sulfonamides, tetracyclines and amphenicols [53]. Similarly, the latest monitoring programme developed by the EFSA reports occasional presence of some of these drug residues in milk, including NSAIDs (diclofenac, salicylic acid or flunixin), antibiotics (fluoroquinolones, macrolides) or antiparasitic drugs (triclabendazole) [54]. Human medicine must also be regulated to prevent drug exposure in suckling infants with medicated mothers. In these cases, it is important to balance the benefits of breastfeeding and the risks of drug exposure of the infants [55]. The lactation stage is a critical feature when considering neonatal exposure risk, since it affects the milk/plasma ratios of some actively secreted drugs [56]. Nonetheless, every drug must be proved to be safe before being administered during lactation.

Animals and humans are also exposed to a large number of environmental contaminants, including mycotoxins, industrial pollutants, carcinogens or pesticides. These compounds may be absorbed by ingestion, inhalation or dermal contact and subsequently concentrated into breast or ruminant milk, causing deleterious effects in potential consumers [57,58].

To avoid the damage induced by all these harmful compounds, apart from ABC/SLC transporters, most tissues and organs are provided with different metabolizing systems. Xenobiotic metabolism is usually performed in three phases: an oxidative process mediated by cytochrome P450 (CYP) superfamily (phase I), a conversion of hydrophobic compounds into hydrophilic derivatives through conjugation (phase II) and excretion by transporters (phase III) [59]. Thus, CYP enzymes are necessary for normal development in mammals and their wide substrate specificity supports their multiple roles in the metabolism of endogenous and exogenous compounds. Although their activity has been mainly detailed in liver, some CYPs are also expressed in extrahepatic tissues. For example, CYP1A1, 1B1, 2C, 2D6, 2E1 and 3A4/5 mRNA and/or protein are expressed in the mammary gland from human samples [60,61,62]. There, they may contribute to the clearance and bioactivation of their substrates, including carcinogens such as 7,12-dimethylbenz(a)anthracene [63]. The interplay between ABC/SLC transporters and CYPs constitutes a complex mechanism that works as a protective barrier against xenobiotic compounds, affecting their bioavailability and toxicology. In fact, some drugs and toxins can modulate either CYP or transporter activities by signalling pathways that involve common nuclear receptors, which results in the enhancement of detoxification and homeostatic cell survival response [59,64,65]. Moreover, some compounds have been described as common substrates for both CYPs and transporters. This is the case for the mycotoxin zearalenone, identified as an ABCG2 substrate [66] but also as a modulator of CYP1A1 and CYP1B1 activity in a breast cancer cell line [67]. Interestingly, drug-drug interactions between CYP and transporter substrates can lead to unpredictable pharmacokinetic alterations [68]. Although this coordinated mechanism has been well characterized in liver or intestine [69,70,71], there are still hardly any studies on this relationship in mammary gland.

This review will focus on SLC and ABC transporters with increased expression in the mammary gland during lactation and their role in the transfer of compounds into milk, including ingredients with high nutritional interest such as vitamins, peptides, neuroactive compounds or steroids, as well as potentially harmful products such as drugs or contaminants. Additional factors, such as regulation, polymorphisms or drug-drug interactions will also be described.

## 2. Transporters with Increased Expression in Lactating Mammary Gland

### 2.1. Influx Transporters: SLCs

#### 2.1.1. OATPs

OATPs belong to the SLCO superfamily (formerly SLC21A). In lactating mammary epithelium there is an increased expression of the OATPs OATP1A2 (encoded by *SLCO1A2*) and OATP2B1 (encoded by *SLCO2B1*). OATP2B1 is also widely expressed throughout the body, including in the myoepithelium surrounding ductal epithelial cells in human mammary gland [72,73,74]. The SLCO superfamily includes sodium-independent transport systems that mediate the transport of a broad range of endobiotics and xenobiotics. Substrates are mainly amphipathic organic anions with a molecular weight of over 300 Da but some of the known transported substrates are also neutral or even positively charged.

Retinol (vitamin A) has been described as an in vitro substrate of OATP1A2, participating in the uptake of retinol in the human retinal pigmented epithelium [75]. Since the placental transport of retinol from the mother to the foetus is blocked during the first stages of pregnancy, the retinol content in milk is critical, especially for preterm newborns, whose feeding depends exclusively on this product [76]. However, until now, it remains unclear whether OATP1A2 is also involved in transepithelial transfer of retinol in the mammary gland.

The steroid hormone conjugates estrone-3-sulfate (ES) and dehydroepiandrosterone sulfate (DHEAS) can be taken up either by OATP1A2 or OAT2B1, as demonstrated using in vitro models [74,77,78]. Interestingly, this transport can be modulated by free steroids. Thus, OATP2B1 uptake of ES, DHEAS or pregnenolone-sulfate can be inhibited by the androgen testosterone or enhanced by gestagens such as progesterone [79]. However, involvement of these transporters in steroid disposition in the mammary gland has not been specifically reported.

Both OATP1A2 and OATP2B1 are also involved in the uptake of small peptides, such as the neuroactive peptides substance P and vasoactive intestinal peptide [80]. Other endogenous OATP substrates include bile acids, thyroid hormones or prostaglandins [81,82,83]. Nevertheless, whether OATPs transport these specific substrates in the mammary gland is still unknown.

Furthermore, among other well characterized substrates of OATPs there are numerous exogenous compounds, including drugs such as statins, angiotensin-converting enzyme inhibitors, angiotensin receptor blockers, antibiotics, antihistamines, antihypertensives and anticancer drugs [46] and the mycotoxin ochratoxin A [84]. Implication of these transporters in their transfer into milk has not been proved.

#### 2.1.2. OCTs

OCTs are classified in the SLC22A superfamily. The OCTs upregulated in mammary gland during lactation are OCT1 (encoded by *SLC22A1*) and OCTN1 (encoded by *SLC22A4*). OCTs transport mainly organic cations. In addition to endogenous substrates, such as steroids, hormones and neurotransmitters, numerous drugs and other xenobiotics are transported by these proteins [85].

The presence of thiamine (vitamin B_1_) in milk depends on the activity of OCT transporters, as reported by Kato et al. [86] who found a dramatically decreased milk/plasma ratio of this vitamin in wild-type compared to *Oct1/2* double-knockout mice. Since double-knockout mice were used in this study, the results obtained could be attributed either to Oct1 or Oct2 activity. However, since Oct2 is not expressed in the mammary gland [18], Oct1 seems to be responsible for thiamine uptake in the mammary gland.

OCTs are also called non-neuronal monoamine transporters since they play a role in the reuptake of biogenic amines, which are important in neurotransmission or cell signalling. In particular, Oct1 is involved in the uptake of several monoamines, including adrenaline, noradrenaline, dopamine, serotonin and tyramine, as demonstrated using cell culture models overexpressing the rat variant of the transporter [87,88]. OCT1 also participates in the uptake of polyamines such as spermidine but with much lower affinity compared to the prototypic OCT1 substrate 1-methyl-4-phenylpyridinium (MPP^+^) [89]. Another neurotransmitter, acetylcholine, has been suggested as an OCT1 substrate since its transport in human placental villus was reversed using OCT1 inhibitors [90]. This hypothesis was later confirmed using oocytes overexpressing both rat and human OCT1 [91]. Therefore, OCT1 may participate in the uptake of these compounds in the mammary gland, although, to the best of our knowledge, there are no specific studies on this tissue.

Whether OCTs are capable of transporting prostaglandins (which are anionic compounds at physiological pH) is controversial. On the one hand, uptake of the prostaglandins E_2_ and F_2α_ was higher in cells overexpressing human OCT1 and OCT2 than in control cells [92]. However, Harlfinger et al. [93] failed to reproduce these results in OCT2 overexpressing cells and they consider that the results obtained in the previous study reflected prostaglandin binding rather than transport.

OCT1 may also be involved in the uptake of nucleosides and their analogues, as reported in oocytes overexpressing the rat variant of the transporter [94].

Excretion of cationic drugs such as cimetidine, acyclovir and nitrofurantoin into milk is higher than expected from simple diffusion [95,96,97]. Using *Abcg2* knockout and *Oct1/2* double-knockout mice, Ito et al. [23] demonstrated that milk transfer of cimetidine and acyclovir required not only Abcg2 but also Oct1/2 expression. This study confirmed cooperative vectorial transport in the mammary gland involving OCTs as uptake and ABCG2 as efflux transporters. Additionally, milk secretion of the bronchodilator terbutaline and the antidiabetic metformin, was decreased in *Oct1/2* double-knockout [23].

OCTN1 shows a high sequence homology with OCTN2, which transports l-carnitine into cells [98,99]. The transport and provision of l-carnitine into milk is important for normal growth and development of the suckling infant [100]. Functional studies have postulated the presence of a carnitine transporter at the mammary epithelium [20,101]. However, the affinity of OCTN1 for l-carnitine is very low compared to that of the antioxidant vitamin ergothioneine, which has been proposed as the physiological substrate of OCTN1 [102]. The selectivity of OCTN1 remains under debate. On the one hand, some reports suggest that this transporter may translocate a wide variety of compounds including nucleoside analogues [103]. However, a recent study failed to find this association and stated that only ergothioneine structurally related compounds such as l-carnitine, tetraethylammonium or gabapentin could be OCTN1 substrates [104]. Considering that OCTN1 expression increases during lactation, additional studies should elucidate its physiologic role in the mammary gland.

#### 2.1.3. CNTs

Something similar occurs with CNTs, which include the three isoforms CNT1, CNT2 and CNT3 (encoded by *SLC28A1*, *SLC28A2* and *SLC28A3*, respectively). These are sodium-dependent symporters, involved in the uptake of nucleosides. Their expression is up to 4-fold higher in lactating versus non-lactating human MEC [18], suggesting a possible role in the transport of nucleosides in the mammary gland. Nucleosides are essential for nucleic acid synthesis and participate in several physiological processes, including cell signalling or acting as energetic metabolites. Although the three members of the CNT family are similar, they show some differences in their substrate specificities: CNT1 shows higher specificity for pyrimidines and, at lower flux rates, also adenosine, CNT2 for purines and CNT3 for both nucleoside classes, while uridine can be taken up by any of the three transporters [105]. Moreover, several nucleoside analogues used in anticancer and antiviral therapy are also CNT substrates. Selectivity for the different nucleoside-based drugs is different for each transporter, CNT3 showing the broadest range of substrates, including purine (gemcitabine, zidovudine or fluorouridines), pyrimidine (ribavirin or clofarabine) and even nucleobase derivatives (6-mercaptopurine or 6-thioguanine) [106].

#### 2.1.4. PEPTs

PEPTs, which belong to the Major Facilitator superfamily (MFS), are high-affinity type proton-coupled peptide transporters. PEPT2 (encoded by *SLC15A2*) is expressed in mammary gland epithelia, where it may contribute to the reuptake of short-chain peptides derived from hydrolysis of milk proteins secreted into the lumen. Shennan et al. [107] observed that the transport of hydrolysis-resistant dipeptides in a perfused rat mammary gland was not inhibited by the excess of competing dipeptides, suggesting that peptide uptake from the circulation was mediated by low affinity mechanisms rather than by PEPT2. Lack of PEPT2 expression in the basolateral membrane of the mammary gland supports this hypothesis [17]. On the other hand, considering the apical expression of the transporter along with its ability to transport several di- and tripeptides, PEPT2 could play a role in the reuptake of oligopeptides from milk back into the epithelial cells [17]. By this mechanism, those peptides secreted into milk, or hydrolysed by milk proteases, may be reabsorbed to serve as an amino acid source. Since PEPT2 also transports a variety of drugs, such as selected β-lactams, angiotensin-converting enzyme inhibitors and antiviral and anticancer metabolites, efficient reabsorption by this mechanism may reduce the burden of xenobiotics in milk [17].

#### 2.1.5. SVCTs

SVCTs belong to the sodium-dependent ascorbic acid transporter family SLC23 [108]. The isoform SVCT2 (namely NCBT1, encoded by *SLC23A2* but formerly *SLC23A1*) is overexpressed in the mammary gland in the final stage of lactation. In contrast to various other mammals, humans are not capable of synthesizing ascorbic acid from glucose and therefore the uptake of ascorbic acid from the diet via SVCT2 is essential for maintaining appropriate concentrations of vitamin C in the human body. This transporter shows a very high affinity especially for the l-ascorbic acid isomer and mediates its uptake by a mechanism dependent on sodium electrochemical gradient [109]. Although its role has been characterized in depth mostly in other tissue models, nothing is known of its role in the mammary gland.

### 2.2. Efflux Transporters: ABCs

ABC transporters efflux various substrates including metabolites, lipids and drugs across cellular membranes by the hydrolysis of ATP. They play a significant role in substrate transport by modifying absorption, distribution and excretion of both drugs and natural compounds and are implicated in lipid and cholesterol transport in many tissues, including the mammary gland [110,111,112]. The main ABC transporters with an increased expression during lactation in mammary gland are ABCC5 and ABCG2.

ABCC5 acts as an efflux transporter of cyclic nucleotides. Since its expression is upregulated in lactation, it may participate in regulating the tissue levels of these signal mediators in the mammary gland, although this fact has not been proved. cAMP and cGMP were the first cyclic nucleotides described as physiological substrates of ABCC5 in vesicle transport studies [113]. Later experiments confirmed that cUMP and cCMP were also effluxed by ABCC5 [114]. A further characterization of this transport in whole cells revealed a low affinity interaction between the transporter and these substrates, so the role of ABCC5 in the modulation of the intracellular levels of cyclic nucleotides may be relevant only under specific conditions (including decreased phosphodiesterase activity or guanylyl cyclase inhibition) [115]. ABCC5 may also be involved in the transfer of hyaluronan to the extracellular matrix, a transport which may be regulated by cGMP levels [116].

The presence of the ABCG2 transporter in the mammary gland has an important role in the transfer of nutrients, drugs and xenobiotics into milk [112]. Murine Abcg2 mediates the secretion of riboflavin (vitamin B_2_) into milk, as demonstrated in an experiment conducted on lactating mice, where the concentration of this vitamin in the milk of wild-type mice was much higher than in *Abcg2-*knockout mice [117]. In this study, the interaction between riboflavin and human ABCG2 was also demonstrated in vitro, so this transporter may have a role in the transfer of riboflavin from the mother to the newborn in humans. Interestingly, pups fed with milk from *Abcg2* knockout mice did not develop clinical symptoms of riboflavin deficiency. Authors suggest a possible compensatory mechanism involved in the transfer of riboflavin equivalents into milk, since flavin adenine nucleotide (FAD) concentration was independent of Abcg2 expression [117].

Biotin (vitamin B_7_) could also be effluxed by ABCG2. The above-mentioned study conducted by van Herwaarden et al. [117] detected a slightly, but significantly, lower concentration of biotin in Abcg2 knockout than in wild-type mice.

Folic acid (vitamin B_9_), essential for a correct neural development and haematological function, has also been described as an ABCG2 substrate. Interaction between this vitamin and drug transporters was initially suspected because MCF7/MX *ABCG2*-overexpressing cells were resistant to the antimetabolite methotrexate [118]. Under this assumption, subsequent in vitro experiments demonstrated that both folic acid and methotrexate were substrates of ABCG2 [119]. However, the contribution of ABCG2 to milk secretion of folate could not be confirmed in vivo, since no differences were found in the milk from wild-type and Abcg2-knockout mice [19,117]. Interestingly, folic acid has also been described as an in vitro substrate of ABCC5, also expressed in the mammary gland [120].

Although vitamin K deficiency is uncommon, it is worth mentioning due to its clinical impact on several biological processes, including blood clotting. The interaction between vitamin K complex (which includes the natural forms phylloquinone or K_1_ and or K_2_ and the synthetic menadione or vitamin K_3_) and ABCG2 has been elucidated. On the one hand, van Herwaarden et al. [117] did not find differences in phylloquinone concentration in the milk between wild-type and *Abcg2*-knockout mice. Nevertheless, the specific interaction between menadione and the substrate binding site of ABCG2, as well as the increased resistance to menadione showed by *ABCG2*-overexpressing cells, suggests that this vitamin is a substrate of this transporter [121]. Since menadione acts as a precursor that can be converted into phylloquinone, it is important to consider the whole complex rather than its individual members.

ABCG2 is also involved in the transport of steroid compounds, including oestrogen and androgen metabolites. In particular, ABCG2 shows a preferential transport of sulfated derivatives such as ES, 17β-oestradiol-sulfate or DHEAS as demonstrated in *ABCG2*-overexpressing cell models [122,123]. Interestingly, experiments conducted on *ABCG2/OATP2B1*-double transfected cells showed increased transport of both ES and DHEAS, thus suggesting coupled activity of both transporters [124]. The glucuronidated steroids 17β-oestradiol-glucuronide, estrone-glucuronide, estradiol-3-glucuronide, estriol-3-glucuronide and estriol-16α-glucuronide have also been described as in vitro ABCG2 substrates, although with much lower affinities than that observed for sulfated derivatives according to their K_m_ values [123,125]. However, these results have not been reproduced in vivo, since the milk secretion of DHEAS did not decrease in *Abcg2-*knockout mice [19].

ABCG2 may also mediate the secretion of bile acids across the mammary gland, since *Abcg2-*knockout mice showed lower levels of these compounds in milk than wild-type mice [126]. Although this pattern was also observed in serum, the differences obtained in milk were maintained even after the administration of taurocholic acid, injected to reach similar serum bile acid levels in both mice strains, thus confirming that the concentrations obtained in milk were not simply a reflection of serum [126].

Moreover, natural compounds present in the diet could be actively secreted into milk by ABCG2. Enterolactone and enterodiol are products obtained from the microbial metabolism of lignans, dietary compounds with estrogenic and antioxidant activities [127]. ABCG2 is relevant in the concentration of these products into milk, since *Abcg2-*knockout mice revealed lower milk/plasma ratios of both compounds than wild-type mice [128,129].

ABCG2 is probably the main contributor to the transfer of drugs into milk. Interaction between this transporter and fluoroquinolones, a class of antibiotics widely used in both human and veterinary medicine, is one of the most studied examples. Ciprofloxacin, ofloxacin and norfloxacin were the first members of the fluoroquinolone family to be identified as substrates of ABCG2 both in vitro and, in the case of ciprofloxacin, in vivo [130]. Involvement of Abcg2 in the transfer of fluoroquinolones into milk has been confirmed in mice, since administration of ciprofloxacin and danofloxacin revealed higher milk/plasma ratios (up to two-fold) in wild-type than in *Abcg2* knockout mice [130,131]. Since several drugs have been described as ABCG2 inhibitors, their co-administration with ABCG2 substrates may modulate the transfer of these antibiotics into milk. In fact, the co-administration of fluoroquinolones with the antiparasitics albendazol-sulfoxide [132] or ivermectin [131] in sheep, reduced milk secretion of enrofloxacin and danofloxacin, respectively. Moreover, apart from drugs, dietary compounds may also affect milk secretion of fluoroquinolones mediated by ABCG2. Administration of the isoflavone genistein decreased milk secretion of enrofloxacin in sheep [132], whereas danofloxacin concentration in sheep milk was lower in animals fed with soy- or flaxseed-enriched diets [133,134].

Fluoroquinolones are not the only antibiotics transported by ABCG2. Nitrofurantoin was identified as an ABCG2 substrate both in vitro and in vivo by Merino et al. [135]. The higher milk/plasma ratio observed in wild-type mice compared to *Abcg2*-knockout animals confirms the role for the transporter in the secretion of nitrofurantoin into milk [135]. These results were later reproduced using a rat “chemical knockout” model based on the co-administration of the antibiotic and the specific Abcg2 inhibitor GF120918 [136]. Regarding ruminants, nitrofurantoin secretion into sheep milk was reduced with the co-administration of the isoflavones genistein or daidzein (which are in vitro inhibitors of ABCG2), either exogenously or through the diet [137].

Similar results have been obtained for some antiparasitic drugs. A high milk/plasma ratio of monepantel-sulfoxide was observed after administration of the anthelmintic monepantel to dairy cows. This metabolite was confirmed as an in vitro bovine ABCG2 substrate, thus suggesting that this transporter is responsible for this metabolite transfer into cow’s milk [106]. Moxidectin, another antiparasitic drug belonging to the macrocyclic lactone family, has also been described as a substrate of ABCG2, based on the observation of a lower milk/plasma ratio in *Abcg2-*knockout mice than in wild-type mice [138]. Milk transfer of moxidectin in sheep decreased after its co-administration with triclabendazol, another antiparasitic drug that inhibits ABCG2 [139].

Information about the transfer of NSAIDs into milk is scarce. Although some of these drugs, such as diclofenac, are in vitro substrates of ABCG2 [140], there is no information about the specific involvement of this transporter in its secretion into milk. Until now, flunixin and its metabolite 5-hydroxyflunixin are the only NSAIDs which have been shown to be transferred into milk by murine and bovine ABCG2 in vivo [141].

Other drugs which have revealed higher milk/plasma ratios in wild-type than in *Abcg2-*knockout mice include the anticancer drug topotecan, the antiviral acyclovir or the antiulcerative cimetidine [19], their Abcg2-mediated secretion into milk having been confirmed. Milk transfer of another antiulcerative drug, pantoprazole, was reversed in rats using the Abcg2 specific inhibitor GF120918 and thus the involvement of this transporter in drug secretion was also confirmed. Interestingly, milk/plasma ratio for (−) pantoprazole was almost 3-fold that of (+) pantoprazole, thus showing that Abcg2 interacts stereoselectively with the two isomers of pantoprazole [142].

Mycotoxins are secondary metabolites produced by several fungi species which can contaminate food products due to fungal infections in crops. These toxins may enter the food chain through the ingestion of polluted food and feed, which is relevant because some of these products have tumorigenic effects or cause liver damage, among many other health issues. Aflatoxin B1, produced by *Aspergillus* spp., has been described as an in vitro substrate of murine and human ABCG2 and its transfer into milk has been confirmed using *Abcg2* knockout mice [143]. This protein is also involved in the transport of enniatins, beauvericin or zearalenone, which are mycotoxins produced by *Fusarium* spp. In particular, the cytotoxicity mediated by enniantins and beauvericin decreased in *ABCG2*-overexpressing cells, possibly due to a higher efflux out of the cell, a hypothesis reinforced by the fact that these toxins have demonstrated specific binding to the transporter [144]. In addition, the estrogenic mycotoxin zearalenone was first described as an ABCG2 substrate by Xiao et al. [66], who observed decreased intracellular accumulation as well as increased resistance to the cytotoxicity mediated by this compound in *ABCG2*-overexpressing cells. The same research group later confirmed the specific involvement of murine and human ABCG2 in bidirectional transport assays and revealed that foetal exposure to this mycotoxin was increased in *Abcg2-*knockout mice [145]. Murine and human ABCG2 is also involved in the efflux of the mycotoxin ochratoxin A in cell culture models [84]. Of all these compounds, milk transfer mediated by Abcg2 of aflatoxin B1 is the only one which has been confirmed in vivo [143], so further studies should be conducted for the other mycotoxins.

Heterocyclic amines are dietary carcinogenic compounds produced during protein heating, so they can be found in overcooked food products or cigarettes. Some of these compounds include 2-amino-1-methyl-6-phenylimidazo(4,5-b)pyridine (PhIP), 2-amino-3-methylimidazo(4,5-f)quinoline (IQ) and 3-amino-1,4-dimethyl-5H-pyrido[4,3-b]indole (Trp-P-1). PhIP, IQ and Trp-P-1 are transferred into milk by the murine Abcg2 as observed in *Abcg2-*knockout mice, although only PhIP and IQ were efficiently transported by the human ABCG2 in vitro [19,143,146].

The endocrine disruptors bisphenol A and perfluorooctanoic acid as well as the organophosphate chlorpyrifos, have been suggested as ABCG2 substrates in cellular models [147,148,149]. Therefore, ABCG2 could have a potential role in the transfer of these harmful compounds into milk.

Table 1 summarizes which compounds are transferred into milk by SLC and ABC transporters according to in vivo studies. However, the in vitro studies conducted until now could be valuable for predicting the behaviour of these transporters in the mammary gland, considering their increased expression during lactation in this tissue. Nevertheless, experimental in vivo studies are needed to confirm this hypothesis.

## 3. Polymorphisms in Transporters with Increased Expression in Lactating Mammary Gland: Potential Effect on Breast-Fed Infants and Dairy Consumers

Expression levels and activities of SLC and ABC transporters may be dependent on their respective genetic variants. Consequently, the study of pharmacogenetics/pharmacogenomics as a covariate in the evaluation of exposure in breast-fed infants and dairy consumers acquires importance. Expression of polymorphic variants of these transporters in the mammary gland may be responsible for the differential transfer of their substrates into milk, thus affecting its quality and composition.

### 3.1. OATP1A2 and OATP2B1 Polymorphisms

Impaired function and expression for variants of human OATP1A2 were observed which could affect the disposition of endogenous compounds and drugs in multiple tissues [154]. In relation to human OATP2B1 genetic variants, it was observed that the R312Q polymorphism decreased plasma concentration of Montelukast [155], frequently used for the treatment of asthma. In addition, the human variant S486F decreased plasma concentration of the antihistamine fexofenadine [156] and the beta-blocker celiprolol [157]. It cannot be ruled out that the differences in the systemic distribution of drugs mediated by the polymorphic variants of OATPs may also affect drug disposition in the mammary gland.

### 3.2. OCT1 and OCTN1 Polymorphisms

Several groups have reported polymorphic variants in the OCT families. In fact, 25 single nucleotide polymorphisms (SNPs) of human OCT1 were identified as affecting transport activity [158,159]. Important differences in transport of important drugs such as morphine [160], metformin [161], lamivudine [162] and ranitidine [163] were observed for genetic polymorphisms of human OCT1. Something similar occurs with human OCTN1 whose polymorphisms affect the transporter of imatinib [164]. Although no evidence for the role of these polymorphic variants in the secretion of drugs into milk was reported, due to the important role of OCT1 in the transfer of compounds into milk, they cannot be ruled out.

### 3.3. PEPT2 Polymorphisms

Numerous SNPs in the gene encoding the human PEPT2 have been described. Haplotype analysis of PEPT2 demonstrates that some SNPs can alter the transport of the model substrate glycyl-sarcosine, which may indicate an alteration in the disposition of drugs transported by PEPT2 [165]. Moreover, the non-synonymous polymorphism R57H disrupts PEPT2 function [166]. Expression of PEPT2 in the epithelial cells of the mammary gland could provide an efficient mechanism for reuptake of short-chain peptides and peptide-based drugs, thereby changing the concentration in milk. Further studies should elucidate whether the polymorphic variants of this transporter affect drug presence in milk.

### 3.4. ABCG2 Polymorphisms

Of all the human polymorphisms of ABCG2 that affect the pharmacokinetics of several drugs, it is important to highlight the role of the SNP Q141K, since it affects response to therapy and clinical outcomes and is associated with diseases such as gout [167]. However, the main reason the polymorphism ABCG2 Q141K stands out in this review is because it is the only human ABCG2 polymorphism reported with an effect on drug secretion into milk. In fact, milk concentration of nifedipine, an ABCG2 substrate, was three times higher in the heterozygous C421A in comparison with C421C women [153].

Cohen-Zinder et al. [168] described a non-synonymous SNP at aminoacidic position 581 of bovine ABCG2 which belongs to a QTL that modifies protein and fat milk composition. The bovine variant ABCG2 Y581S has been described as an in vitro and in vivo gain-of-function polymorphism with a greater transport capacity [169]. Leaving out breastfeeding, SNPs in this transporter have an important role in the presence of several compounds and drugs in milk intended for human consumption.

ABCG2 plays an important role in uric acid transfer into milk, since its secretion was increased in dairy cow carriers of the Y581S polymorphism [150]. As suggested by the authors, the presence of this compound could affect the redox potential of milk. In addition, the bovine polymorphism Y581S affects enterolactone *s*ecretion into cow milk [150]. Danofloxacin, enrofloxacin and ciprofloxacin are also effluxed into milk by bovine ABCG2 more efficiently in polymorphic Y581S than in wild-type cows [151,152,169]. After administration of flunixin to dairy cows, both the parental drug and its metabolite 5-hydroxyflunixin were secreted into milk and in the case of flunixin, its secretion was increased in carriers of the polymorphism Y581S compared to wild-type cows [141].

## 4. Modulation of the Genes Encoding Transporters with Increased Expression in Lactating Mammary Gland

Mammary gland growth, differentiation and lactogenesis require interplay among many different hormones whose function is to exercise control over transcription profiles in the gland. These modifications of transcription profiles result from changes in the sets of genes expressed in this tissue. The steroid hormones oestrogen and progesterone exert important roles during mammary development [170], nevertheless, prolactin is the main hormone for both induction and maintenance of lactation [171,172]. In addition to this relevant role, prolactin also influences expression of several drug transporter genes. For instance, bovine PEPT2 expression is enhanced by prolactin, along with other lactogenic hormones, including insulin and hydrocortisone [173]. Moreover, the binding of prolactin to its receptor activates the tyrosine kinase Janus kinase-2 (JAK2) and the activator of transcription 5 (STAT5). Specifically, signalling through the JAK2/STAT5 cascade has been demonstrated to be indispensable for the specification, proliferation, differentiation and survival of secretory mammary epithelial cells. Studies in knock-out mice have shown that STAT5 isoforms are central for alveolar development and milk gene expression [174,175]. This activation pathway also has a great relevance in the modulation of the genes of drug transporters. It has been demonstrated that prolactin induces ABCG2 expression in T-47D human breast cancer cells by JAK2/STAT5 [176]. An additional feature in this complex system is that ABCG2 itself influences the proliferation of primary bovine mammary epithelial cells (BMECs) [177].

Other complex signalling networks regulate the expression of transporters. As an example, the uptake of the model dipeptide β-alanyl-l-lysyl-Nε-7-amino-4-methyl-coumarin-3-acetic acid in BMECs by PEPT2 is regulated through the phosphatidylinositol-3-kinase (PI3K) / protein kinase B (Akt) pathway. Another well characterized mechanism is the aryl hydrocarbon receptor (AhR) signalling pathway. Coupling between AhR and its ligands results in the translocation of the complex into the nucleus and its subsequent binding to dioxin response elements (DRE) which are localized in the 5’-untranslated region (5’-UTR) of certain mRNAs [178]. Tan et al. [179] demonstrated that human ABCG2 expression was regulated by AhR through specific binding to DRE regions in the ABCG2 promoter. Some compounds, including toxins, may act as agonists of AhR activating this pathway. In this way, incubation of bovine mammary epithelial BME-UV cell line with 2,3,7,8-tetrachlordibenzo-p-dioxin (TCDD) and prochloraz increased ABCG2 gene expression as well as efflux activity [180]. Similarly, Manzini et al. [181] suggested that the dioxin-like PCB may also increase the transport activity of ABCG2 through the same pathway. Considering these previous findings, AhR agonists may enhance milk transfer of chemicals as a result of the induction of ABCG2. Interestingly, AhR ligands can also induce the expression of CYP1A1 and CYP1B1, as demonstrated in rat and human MEC [182,183], a finding that could support a possible interplay between CYPs and transporters such as ABCG2 in the mammary gland.

It is not only hormone control that is important in mammary glands, since the expression of milk proteins, including drug transporters may also be modulated by epigenetic mechanisms. Epigenetic variations of the genes encoding transporters can modulate uptake and excretion of many drugs in the mammary gland. There are numerous reports of milk protein genes that are dramatically regulated during lactation due to epigenetic mechanisms [184]. In the particular case of transporters, several studies have suggested that many genes are under epigenetic control and are being responsible for variations in drug responses [185,186]. The main mechanisms of epigenetic control are DNA-methylation and histone post-translational modifications. DNA methylation and histone acetylation/deacetylation balance is important in milk protein gene expression and in mammary epithelial cell differentiation [187]. Although there is no evidence of the influence of these epigenetic mechanisms on drug transporter expression in the mammary gland, several studies have reported a relevant role in other tissues. Hyper- or hypomethylation at the 5-position carbon of cytosine within 5’-CpG-3’ dinucleotide residues located mainly in promoter region have been shown to repress or increase, respectively, gene expression of drug transporters [186,188]. The methylation status of ABCG2 has been studied in detail. It has been shown that an in vitro methylation of the ABCG2 promoter reduced transcriptional activity [189]. In consequence, hypomethylation status of the ABCG2 promoter produced the overexpression of ABCG2. This fact has been observed in several drug-resistant cancer cell lines [190,191,192]. Schaeffeler et al. [193] suggested that the epigenetic silencing of OCT1 was produced in hepatocellular carcinoma. Concerning histone post-translational modifications, exposure to valproate increased acetylation of histone H4 in the ABCG2 promoter in CMK cells as well as ABCG2 mRNA levels [194]. A recent study demonstrated that treatment with histone deacetylase elicits brain region-specific enhancement of histone H3 acetylation and upregulation of ABCG2 transporter in mice [195].

MicroRNAs (miRNAs) constitute another well-known post-transcriptional epigenetic mechanism. Wang et al. [196] suggested that miRNAs might play a role as regulators of signalling pathways, metabolic enzymes and transporters, affecting milk quality and milk secretion during mammary gland differentiation. Although the specific interaction between miRNAs and transporter genes in the mammary gland is still unknown, it has been clearly demonstrated in other tissues [197,198,199]. Several studies have demonstrated the implications of different miRNAs in regulation of the ABCG2 transporter [199,200]. For instance, To et al. [201] identified a miR-519c binding site in the ABCG2 3’-UTR involved in decreasing endogenous ABCG2 mRNA and protein levels. Although no data have been reported on miRNA-dependent regulation in the OCT1 gene, several in silico studies suggest an evident correlation between the length of 3’-UTR of OCT1 mRNA and regulatory miRNAs such as has-miR-3169 [202].

Emerging evidence has demonstrated that infection and inflammation in various tissues affect the expression and function of transporters with an impact on drug distribution and efficacy of therapy [203,204]**.** Many of reported data are mainly from liver, intestine, kidney, brain and placenta but this effect is important also in the mammary gland. Inflammatory processes have an effect on expression of drug transporters in the mammary gland affecting the availability of nutrients in milk. It has been demonstrated that lipopolysaccharide (LPS) causes a marked decrease in rat Octn1 transporter mRNA expression during lactation [205]. Moreover, *Staphylococcus aureus* infection decreases expression of murine Abcg2 during mastitis, which may affect secretion of drugs into milk and efficacy of drug therapy [206]. Further studies should clarify whether these changes may lead to reductions in transporter substrate availability, including nutrients and drugs, which may significantly impact milk quality and yield. According to this hypothesis, some studies revealed that milk concentration of flunixin and 5OH-flunixin after oral administration of the drug was different in dairy cows with mastitis compared to healthy controls [207,208]. Since both compounds have been described as bovine ABCG2 substrates [141], the differential transport may be due to potential transporter expression changes observed during infection [206].

Interestingly, not only pathogenic microorganisms but also microbiome can affect the expression of transporters. The depletion of the gut microbiota in rats after antibiotic treatment resulted in a decreased expression of Oct1 in the liver, which in turn affected the pharmacokinetics of metformin [209]. Similarly, mice inoculated with *Lactobacillus ingluviei* as probiotic treatment, showed increased mRNA levels of ABCG2 compared to the control group [210]. An extensive analysis of germ-free and antibiotic-treated mice revealed that intestinal flora affected liver and kidney expression of both CYPs and transporters, including Oct1 and Abcg2 among others [211]. Considering these results, we cannot discard that microbioma could affect the expression of transporters in other extraintestinal tissues, such as the mammary gland, although this hypothesis should be elucidated in future studies

## 5. Conclusions

Without doubt, the lactation process involves important physiological changes, some of which are linked to drug transporters. The function of these transporters during this period is to transfer nutrients and compounds of various types into milk, contributing to its composition and, collaterally, to the presence of drugs, carcinogens and pollutants, which represents a public health as well as a food quality and safety problem. This dual function is the result of the broad range of endobiotics and xenobiotics which are substrates of the transporters with increased expression and activity in lactating mammary gland, belonging to the SLC and ABC families. Complex mechanisms are involved in the regulation of the physiological function and development of the mammary gland as well as in the uptake, reuptake and efflux of a large group of compounds including drugs in this organ. In this context, the efflux of undesirable compounds to milk represents a relevant issue, since breast-fed infants and dairy consumers may be exposed to drugs, mycotoxins and pesticides, which constitutes one of the main causes of food allergies and intolerances, antibiotic resistance, hormonal disturbances and poisoning.

The influence of ABCG2 in the composition and quality of milk has been studied in depth during the last decades. Although this transporter seems to play the most relevant role, other less characterized transporters may also be involved. OCTs represent the main SLC transporters in the mammary gland during lactation as revealed in in vivo studies, where they may contribute to the uptake of thiamine and biogenic amines and to cooperative vectorial transport for some drugs. PEPT2, present in mammary epithelia, may explain the importance of the reuptake of short chain peptides derived from hydrolysis of milk proteins. Although the role of OATPs, CNTs and SVCT2 in the mammary gland has not been completely elucidated, their higher expression during lactation suggests that they may be involved in the uptake of physiological compounds.

This review highlights the importance of acquiring new knowledge of these processes, in order to implement control policies on desired milk quality in animal production and to advise nursing mothers about the risk of transferring unwanted compounds into milk.

## Figures and Tables

**Figure 1 nutrients-11-02372-f001:**
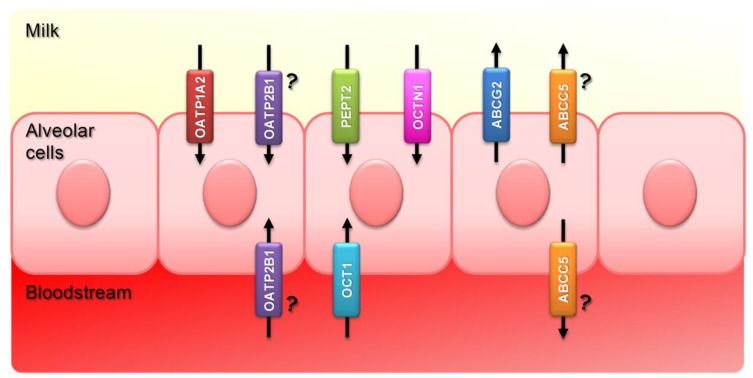
Subcellular localization of the main ABC- and SLC-transporters upregulated in the mammary gland during lactation. The apical localization of ABCG2 and PEPT2 in the mammary gland has been confirmed in previous studies [17,19]. The localization of OATP1A2, OCT1 and OCTN1 suggested in this figure is based on their localization in other tissues [35,36,37,38,39]. OATP2B1 and ABCC5 are localized apically or basolaterally, depending on the tissue [40,41,42,43,44,45,46]; however, their specific localization in the mammary gland is still unclear.

**Table 1 nutrients-11-02372-t001:** Milk transfer of SLC and ABC substrates confirmed in vivo.

Transporter	Substrate	Species
OCT1	Endogenous:	
	Thiamine	Murine ^1^ [86]
	Drugs:	
	Cimetidine	Murine ^1^ [23]
	Acyclovir	Murine ^1^ [23]
ABCG2	Endogenous:	
	Riboflavin	Murine ^1^ [117]
	Biotin	Murine ^1^ [117]
	Bile acids	Murine ^1^ [126]
	Uric acid	Bovine ^2^ [150]
	Dietary:	
	Enterolactone	Murine ^1^ [128,129], bovine ^2^ [150]
	Enterodiol	Murine ^1^ [129]
	Drugs:	
	Ciprofloxacin	Murine ^1^ [130], bovine ^2^ [151]
	Danofloxacin	Murine ^1^ [131], ovine ^3^ [131], bovine ^2^ [152]
	Enrofloxacin	Ovine ^3^ [132], bovine ^2^ [151]
	Nitrofurantoin	Murine ^1^ [135], rat ^3^ [136], ovine ^3^ [137]
	Moxidectin	Murine ^1^ [138], ovine ^3^ [139]
	Flunixin and 5-hydroxyflunixin	Murine ^1^ [141], bovine ^2^ [141]
	Topotecan	Murine ^1^ [19]
	Acyclovir	Murine ^1^ [19]
	Cimetidine	Murine ^1^ [19]
	Pantoprazole	Rat ^3^ [142]
	Nifedipine	Human ^2^ [153]
	Toxins:	
	Aflatoxin B1	Murine ^1^ [143]
	Heterocyclic amines (PhIP, IQ, Trp-P-1)	Murine ^1^ [19,143]

^1^ Substrates confirmed based on the comparison between wild-type and knockout animals. ^2^ Substrates confirmed based on the comparison between wild-type and polymorphic variants of the bovine or human transporter. ^3^ Substrates confirmed based on the comparison between animals with and without co-administration of specific inhibitors of the transporter. ABC, ATP-binding cassette; SLC, Solute Carrier.

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
