# Peer review of "Transporters in the Mammary Gland—Contribution to Presence of Nutrients and Drugs into Milk"

_nutrients, 2019, doi:10.3390/nu11102372_

Round 1

Reviewer 1 Report

Nutrients

Authors (AA): Alba M. García-Lino, Indira Álvarez-Fernández, Esther Blanco-Paniagua, Gracia Merino, and Ana I. Álvarez

Manuscript (MS) title: Transporters in the mammary gland: contribution to presence of nutrients and drugs into milk (nutrients-570755)

Reviewer’s report

In the present MS (a review), the AA provide Readers with general comparative knowledge on drug transporters (DTs), namely the Solute Carrier and the ATP Binding Cassette transporters (SLC and ABC DTs, respectively). More in detail, they discuss about their:

constitutive expression in mammary gland and during lactation regulation role in afflux/efflux of nutrients/xenobiotics

Nutrients is an international, peer-reviewed open access journal publishing studies related to human nutrition. The journal accept also MS describing the outcomes of animal studies that have relevance to human health. It publishes reviews.

Veterinary pharmaco-toxicologists and food toxicologists know that DTs are likely to affect the disposition of xenobiotics and endogenous compounds once entered/within the living organism. This behaviour has undeniably some effects on human health, either directly (e.g., assumption of drugs, exposure to contaminants) than indirectly (eating animal food-products, like meat and milk).

Therefore, this review is of interest for Nutrients. Moreover, to the best of my opinion no review on DTs have been published in this journal. On the contrary, PubMed searches for drug transporters AND mammary gland (review) and drug transporters AND lactation (review) give 24 and 23 articles, respectively.

In my opinion, this review is therefore of interest for Nutrients. Nevertheless, it cannot be accepted for publication in this journal (a Q1-ranked peer-reviewed journal) as it is. A MAJOR REVISION opinion is attributed to the MS.

A list of more detailed major, minor and discretionary comments is hereby reported. The corresponding MS page(s) and line(s) are here indicated as P(P) and L(L).

Major comments

All throughput the MS, but specifically in the Introduction, there is no mention to the known relationships between DTs and drug metabolizing enzymes, e.g. cytochromes P450 (CYPs). This relationship is of greater importance in all the species, because both superfamilies contribute to xenobiotics and endogenous compounds disposition. As an example, mycotoxins are ingested orally and metabolized (bioactivated) by CYPs. But some of them are substrates of SLC/ABC transporters (e.g., ochratoxin A, aflatoxin B1), too. Moreover, some CYPs are also constitutively expressed in mammary gland. Please, add some concepts/knowledge about this issue.

The MS essentially focuses on human and Ruminant DTs; therefore, it is not a true comparative (cross-species comparison) paper on DTs expression, regulation and biological activity. Other animal species are endowed with DTs.

I understand the major importance attributed to dairy products food-producing species, for the obvious consumers’ risk. Moreover, the AA have a good experience on Ruminants DTs.

Taking into consideration the journal, its aims and scope, and the subjects, I think the AA should concentrate their efforts on food-producing species (maybe including pigs, poultry, …) and review the knowledge about DTs in this species, including the potential risks for consumers. This would justify the citation about drugs, contaminants besides nutrients, for the indirect risk of exposure for babies and humans. Furthermore, a number of review papers have been published about DTs in humans, but few (if any) about the relevance of DTs in food-producing species and their role in determining the risk of xenobiotic exposure. In my opinion, this might result in a higher number of citations, in light of the recent improvements in the field of food safety. Obviously, the expression of DTs in humans and neonates (specifically, their role in nutrients absorption) should be maintained in the MS.

This clearly means the AA should hardly reformat the MS and focus on DTs of animal species. I strongly encourage this.

Another challenging point to be added is the relationship between animals DTs (but also drug metabolizing enzymes) and microbiome/microbiota. Few papers have considered this relationship in food-producing species, but a number of papers are available for humans. Ts might be an additive value for this review. Please consider updating.

Maybe differences in composition of milk between Ruminants, neonates (and other animal species, if any/of nterest) might help understanding species-differences in xenobiotic uptake and concentration. Possible Tables might be added to as Supplementary information.

Maybe some paragraph about physio-pathological conditions affecting DTs expression and function might improve the MS relevance.

Minor comments

Section 2

Please, consider adding subsections termed afflux DTs and efflux DTs.

Figure 1 legend

Please, circumstantiate concepts adding references.

P2, LL55-57

When talking about constitutive expression in different tissues, minor fold-changes (compared to control tissue, e.g., liver) are likely to be observed. Nevertheless, the AA cannot affirm that 1.5 fold-changes are higher levels. They should not forget that the threshold for quantitative Real-Time RT-PCR PCR results is usually set at 2.0 fold-changes. Please revise/clarify.

P2, LL61-62

Compared to what tissue? To mammary gland not in lactation? Else? Please, clarify.

P2, LL65-67

Despite the citation ([24]), please add the species compared with Ruminants.

P4, L147

…androgen influence? Please, clarify and add references.

P9, LL377-379

Add specific references.

P9, LL391-396

This sentence is just a bit difficult to be understood. Please consider resentencing.

Editorial comments

References

While checking for cited references, I found some mistakes in citation, e.g. unocrrect page number citation, … . Please, check and revise according to Instructions for Authors.

P1, L35

Proteins (-3.5%), sugars (-7%), …

P2, L51

Consequently, … (? Indeed,?)

P4, L128

I disagree with Transporter induction in … . Maybe .. Increased expression and activity of DTs in mammary gland?

Consider that induction/inhibition ensue transcriptional/post-transcriptional-translational mechanisms of regulation (same question LL253-254).

P8, L327

[101] in sheep, reduced ….

P8, L368

…by Fusarium spp. In …

Reviewer 2 Report

I read with great interest your manuscript entitled “Transporters in the mammary gland: contribution to presence of nutrients and drugs into milk”. The manuscript summarized a great deal of work in the field, in a succinct way. I found it complete in its inclusion of endogenous substrates as well as drugs and xenobiotics. I think the figure and table are important to the manuscript.

I also think the publication of this manuscript is extremely timely in light of new regulations in the USA. The 21st Century Cures Act , mandates the inclusion of women and minorities in clinical trials and created the Task Force on Research Specific to Pregnant and Lactating Women. A report issued by this task force in Sept 2018 is given here: https://www.nichd.nih.gov/sites/default/files/2018-09/PRGLAC_Report.pdf

Specifically, it mentions that “Evidence-based answers are required for women and their clinicians to make fully informed choices based on the risks and benefits of medicating or not medicating conditions during pregnancy and lactation.”

A deeper understanding by the entire field, of the transport processes in the lactating mammary gland is crucial for study design and protection of women and their infants.

I found very few places for edits.

Specific changes:

Line 35: include the species- “Mature human milk”

Line 35: Please clarify if these are vol% or cal% values

Line 38: lysozyme (spelling)

Figure 1 caption: Change “overexpressed” to “upregulated”

Line 123: change “induced” to “increased”

Line 148: change “gestagenes” to “gestagens”

Line 160: Change “overexpressed” to “upregulated”

Line 235: separate into two words “aminoacid”

Line 263: Change “guanilyl” to “guanylyl”

Line 327: missing a p in sheep

Line 405: pharmacogenetics (missing a c)

Lines 443 & 444: be consistent in your nomenclature for SNPs, for clarity

Line 447: gain-of-function

Line 464: prolactin is the main hormone

Line 530-533: Please add reference if possible- mastitis modulates excretion of nutrients…

Round 2

Reviewer 1 Report

I carefully read the new MS version, and I found it greatly ameliorated.